# Learning through Dialogue Interactions by Asking Questions

**Jiwei Li, Alexander H. Miller, Sumit Chopra, Marc'Aurelio Ranzato, Jason Weston**
Facebook AI Research,
New York, USA
`{jiwel,ahm,spchopra,ranzato,jase}@fb.com`

## Abstract

A good dialogue agent should have the ability to interact with users by both responding to questions and by asking questions, and importantly to learn from both types of interaction. In this work, we explore this direction by designing a simulator and a set of synthetic tasks in the movie domain that allow such interactions between a learner and a teacher. We investigate how a learner can benefit from asking questions in both offline and online reinforcement learning settings, and demonstrate that the learner improves when asking questions. Finally, real experiments with Mechanical Turk validate the approach. Our work represents a first step in developing such end-to-end learned interactive dialogue agents.

## 1 Introduction

When a student is asked a question by a teacher, but is not confident about the answer, they may ask for clarification or hints. A good conversational agent (a learner/bot/student) should have this ability to interact with a dialogue partner (the teacher/user). However, recent efforts have mostly focused on learning through fixed answers provided in the training set, rather than through interactions. In that case, when a learner encounters a confusing situation such as an unknown surface form (phrase or structure), a semantically complicated sentence or an unknown word, the agent will either make a (usually poor) guess or will redirect the user to other resources (e.g., a search engine, as in Siri). Humans, in contrast, can adapt to many situations by asking questions.

We identify three categories of mistakes a learner can make during dialogue[1]: (1) the learner has problems understanding the surface form of the text of the dialogue partner, e.g., the phrasing of a question; (2) the learner has a problem with reasoning, e.g. they fail to retrieve and connect the relevant knowledge to the question at hand; (3) the learner lacks the knowledge necessary to answer the question in the first place – that is, the knowledge sources the student has access to do not contain the needed information.

All the situations above can be potentially addressed through interaction with the dialogue partner. Such interactions can be used to *learn to perform better in future dialogues*. If a human student has problems understanding a teacher's question, they might ask the teacher to clarify the question. If the student doesn't know where to start, they might ask the teacher to point out which known facts are most relevant. If the student doesn't know the information needed at all, they might ask the teacher to tell them the knowledge they're missing, writing it down for future use.

In this work, we try to bridge the gap between how a human and an end-to-end machine learning dialogue agent deal with these situations: our student has to *learn how to learn*. We hence design a simulator and a set of synthetic tasks in the movie question answering domain that allow a bot to interact with a teacher to address the issues described above. Using this framework, we explore how a bot can benefit from interaction by asking questions in both offline supervised settings and online reinforcement learning settings, as well as how to choose when to ask questions in the latter setting. In both cases, we find that the learning system improves through interacting with users.

---

[1]This list is not exhaustive; for example, we do not address a failure in the dialogue generation stage.

Finally, we validate our approach on real data where the teachers are humans using Amazon Mechanical Turk, and observe similar results.

## 2 RELATED WORK

Learning language through interaction and feedback can be traced back to the 1950s, when Wittgenstein argued that the meaning of words is best understood from their use within given language games (Wittgenstein, 2010). The direction of interactive language learning through language games has been explored in the early seminal work of Winograd (Winograd, 1972), and in the recent SHRD-LURN system (Wang et al., 2016). In a broader context, the usefulness of feedback and interactions has been validated in the setting of multiple language learning, such as second language learning (Bassiri, 2011) and learning by students (Higgins et al., 2002; Latham, 1997; Werts et al., 1995).

In the context of dialogue, with the recent popularity of deep learning models, many neural dialogue systems have been proposed. These include the *chit-chat* type *end-to-end dialogue systems* (Vinyals & Le, 2015; Li et al., 2015; Sordoni et al., 2015), which directly generate a response given the previous history of user utterance. It also include a collection of *goal-oriented* dialogue systems (Wen et al., 2016; Su et al., 2016; Bordes & Weston, 2016), which complete a certain task such as booking a ticket or making a reservation at a restaurant. Another line of research focuses on supervised learning for question answering from dialogues (Dodge et al., 2015; Weston, 2016), using either a given database of knowledge (Bordes et al., 2015; Miller et al., 2016) or short stories (Weston et al., 2015). As far as we know, current dialogue systems mostly focus on learning through fixed supervised signals rather than interacting with users.

Our work is closely related to the recent work of Weston (2016), which explores the problem of learning through conducting conversations, where supervision is given naturally in the response during the conversation. Their work introduced multiple learning schemes from dialogue utterances. In particular the authors discussed *Imitation Learning*, where the agent tries to learn by imitating the dialogue interactions between a teacher and an expert student; *Reward-Based Imitation Learning*, which only learns by imitating the dialogue interactions which have have correct answers; and *Forward Prediction*, which learns by predicting the teacher's feedback to the student's response. Despite the fact that Forward Prediction does not uses human-labeled rewards, the authors show that it yields promising results. However, their work did not fully explore the ability of an agent to learn via questioning and interaction. Our work can be viewed as a natural extension of theirs.

## 3 THE TASKS

In this section we describe the dialogue tasks we designed[2]. They are tailored for the three different situations described in Section 1 that motivate the bot to ask questions: (1) *Question Clarification*, in which the bot has problems understanding its dialogue partner's text; (2) *Knowledge Operation*, in which the bot needs to ask for help to perform reasoning steps over an existing knowledge base; and (3) *Knowledge Acquisition*, in which the bot's knowledge is incomplete and needs to be filled.

For our experiments we adapt the WikiMovies dataset (Weston et al., 2015), which consists of roughly 100k questions over 75k entities based on questions with answers in the open movie dataset (OMDb). The training/dev/test sets respectively contain 181638 / 9702 / 9698 examples. The accuracy metric corresponds to the percentage of times the student gives correct answers to the teacher's questions.

Each dialogue takes place between a teacher and a bot. In this section we describe how we generate tasks using a simulator. Section 4.2 discusses how we test similar setups with real data using Mechanical Turk.

The bot is first presented with facts from the OMDb KB. This allows us to control the exact knowledge the bot has access to. Then, we include several teacher-bot question-answer pairs unrelated to the question the bot needs to answer, which we call *conversation histories*[3]. In order to explore the

---

[2] Code and data are available at `https://github.com/facebook/MemNN/tree/master/AskingQuestions`.
[3] These history QA pairs can be viewed as distractions and are used to test the bot's ability to separate the wheat from the chaff. For each dialogue, we incorporate 5 extra QA pairs (10 sentences).

benefits of asking clarification questions during a conversation, for each of the three scenarios, our simulator generated data for two different settings, namely, *Question-Answering* (denoted by QA), and *Asking-Question* (denoted by AQ). For both *QA* and *AQ*, the bot needs to give an answer to the teacher's original question at the end. The details of the simulator can be found in the appendix.

## 3.1 QUESTION CLARIFICATION.

In this setting, the bot does not understand the teacher's question. We focus on a special situation where the bot does not understand the teacher because of typo/spelling mistakes, as shown in Figure 1. We intentionally misspell some words in the questions such as replacing the word "movie" with "movvie" or "star" with "sttar".[4] To make sure that the bot will have problems understanding the question, we guarantee that the bot has never encountered the misspellings before—the misspelling-introducing mechanisms in the training, dev and test sets are different, so the same word will be misspelled in different ways in different sets. We present two *AQ* tasks: (i) *Question Paraphrase* where the student asks the teacher to use a paraphrase that does not contain spelling mistakes to clarify the question by asking "what do you mean?"; and (ii) *Question Verification* where the student asks the teacher whether the original typo-bearing question corresponds to another question without the spelling mistakes (e.g., "Do you mean which film did Tom Hanks appear in?"). The teacher will give feedback by giving a paraphrase of the original question without spelling mistakes (e.g., "I mean which film did Tom Hanks appear in") in *Question Paraphrase* or positive/negative feedback in *Question Verification*. Next the student will give an answer and the teacher will give positive/negative feedback depending on whether the student's answer is correct. Positive and negative feedback are variants of "No, that's incorrect" or "Yes, that's right"[5]. In these tasks, the bot has access to all relevant entries in the KB.

## 3.2 KNOWLEDGE OPERATION

The bot has access to all the relevant knowledge (facts) but lacks the ability to perform necessary reasoning operations over them; see Figure 2. We focus on a special case where the bot will try to understand what are the relevant facts. We explore two settings: *Ask For Relevant Knowledge* (Task 3) where the bot directly asks the teacher to point out the relevant KB fact and *Knowledge Verification* (Task 4) where the bot asks whether the teacher's question is relevant to one particular KB fact. The teacher will point out the relevant KB fact in the *Ask For Relevant Knowledge* setting or give a positive or negative response in the *Knowledge Verification* setting. Then the bot will give an answer to the teacher's original question and the teacher will give feedback on the answer.

## 3.3 KNOWLEDGE ACQUISITION

For the tasks in this subsection, the bot has an incomplete KB and there are entities important to the dialogue missing from it, see Figure 3. For example, given the question "Which movie did Tom Hanks star in?", the missing part could either be the entity that the teacher is asking about (question entity for short, which is Tom Hanks in this example), the relation entity (starred_actors), the answer to the question (Forrest Gump), or the combination of the three. In all cases, the bot has little chance of giving the correct answer due to the missing knowledge. It needs to ask the teacher the answer to acquire the missing knowledge. The teacher will give the answer and then move on to other questions (captured in the conversational history). They later will come back to reask the question. At this point, the bot needs to give an answer since the entity is not new any more.

Though the correct answer has effectively been included in the earlier part of the dialogue as the answer to the bot's question, as we will show later, many of the tasks are not as trivial as they look when the teacher reasks the question. This is because the bot's model needs to memorize the missing entity and then construct the links between the missing entities and known ones. This is akin to the real world case where a student might make the same mistake again and again even though each time the teacher corrects them if their answer is wrong. We now detail each task in turn.

---

[4]Many reasons could lead to the bot not understanding the teacher's question, e.g., the teacher's question has an unknown phrase structure, rather than unknown words. We choose to use spelling mistakes because of the ease of dataset construction.

[5]In the datasets we build, there are 6 templates for positive feedback and 6 templates for negative feedback.

**Task 1: Question Paraphrase (QA)**
kb: Larry Crowne directed_by Tom Hanks
kb: Forrest Gump starred_actors Tom Hanks, Robin Wright, Gary Sinise
kb: Forrest Gump directed_by Robert Zemeckis
T/S : Conversation History.
T : Which movvie did Tom Hanks sttar in ?
S : Larry Crowne
T : Wrong. (-)

**Task 1: Question Paraphrase (AQ):**
kb: Larry Crowne directed_by Tom Hanks
kb: Forrest Gump starred_actors Tom Hanks, Robin Wright, Gary Sinise
kb: Forrest Gump directed_by Robert Zemeckis
T/S : Conversation History.
T : Which movvie did Tom Hanks sttar in ?
S : What do you mean ?
T : I mean which film did Tom Hanks appear in.
T : Which movvie did Tom Hanks sttar in ?
S : Forrest Gump
T : That's correct. (+)

**Task 2: Question Verification (QA):**
kb: Larry Crowne directed_by Tom Hanks
kb: Forrest Gump starred_actors Tom Hanks, Robin Wright, Gary Sinise
kb: Forrest Gump directed_by Robert Zemeckis
T/S : Conversation History.
T : Which movvie did Tom Hanks sttar in ?
S : Forrest Gump
T : That's correct. (+)

**Task 2: Question Verification (AQ)**
kb: Larry Crowne directed_by Tom Hanks
kb: Forrest Gump starred_actors Tom Hanks, Robin Wright, Gary Sinise
kb: Forrest Gump directed_by Robert Zemeckis
T/S : Conversation History.
T : Which movvie did Tom Hanks sttar in ?
S : Do you mean which film did Tom Hanks appear in ?
T : That's correct. (+)
T : Which movvie did Tom Hanks sttar in ?
S : Larry Crowne
T : Sorry, that's not right. (-)

Figure 1: Sample dialogues for the two *Question Clarification* tasks (rows) using both the traditional *QA* setting (left column) and *AQ* setting (right column). In each case the same example is given for simplicity. Black text prefixed by "kb:" denotes KB knowledge that the student has access to. Blue text is spoken by the teacher, denoted by T. (+) denotes a reward of 1 (and 0 otherwise) that the teacher assigns to the bot. Red text denotes responses or questions posed by the bot, denoted by S. Brown denotes typos deliberately introduced by the authors. For the *Question Verification* setting, the student can either ask a correct (pertinent) question (as in this example) or an incorrect (irrelevant) one. The teacher will give positive or negative feedback based on the correctness of the student's question. In our offline superised learning experiments, the probability of asking pertinent questions and correctly answering the original question from the teacher is set to 0.5. Finally, T/S denotes 5 pairs of questions and answers that are irrelevant to the rest of the conversation.

**Task 3: Ask For Relevant Knowledge (AQ):**
kb: Larry Crowne directed_by Tom Hanks
kb: Forrest Gump starred_actors Tom Hanks, Robin Wright, Gary Sinise
kb: Forrest Gump directed_by Robert Zemeckis
T/S : Conversation History.
T : Which movie did Tom Hanks star in ?
S : Can you give me a hint ?
T : It is related to the kb fact that
    Forrest Gump starred_actors Tom Hanks
    Robin Wright, Gary Sinise
T : Which movie did Tom Hanks star in ?
S : Forrest Gump
T : That's correct. (+)

**Task 4: Knowledge Verification (AQ):**
kb: Larry Crowne directed_by Tom Hanks
kb: Forrest Gump starred_actors Tom Hanks, Robin Wright, Gary Sinise
kb: Forrest Gump directed_by Robert Zemeckis
T/S : Conversation History.
T : Which movie did Tom Hanks star in ?
S : Does it have something to do with
    Larry Crowne directed_by Tom Hanks?
T : No. That's wrong. (-)
T : Which movie did Tom Hanks star in ?
S : Forrest Gump
T : That's correct. (+)

Figure 2: Sample dialogues for *Knowledge Operation* tasks.

**Missing Question Entity**: The entity that the teacher is asking about is missing from the knowledge base. All KB facts containing the question entity will be hidden from the bot. In the example for *Task 5* in Figure 3, since the teacher's question contains the entity Tom Hanks, the KB facts that contain Tom Hanks are hidden from the bot.

Figure 3: Different Tasks for **Knowledge Acquisition**. Crossed lines correspond to entries of the KB whose retrieval is missed.

.

**Task 5: Missing Question Entity (QA)**
~~kb: Larry Crowne directed_by Tom Hanks~~
~~kb: Forrest Gump starred_actors Tom Hanks~~
kb: Forrest Gump starred_actors Sally Field
kb: Forrest Gump directed_by Robert Zemeckis
T/S : Conversation History.
T : Which movie did Tom Hanks star in ?
S : Forrest Gump
T : That's correct. (+)

**Task 5: Missing Question Entity (AQ)**
~~kb: Larry Crowne directed_by Tom Hanks~~
~~kb: Forrest Gump starred_actors Tom Hanks,~~
kb: Forrest Gump starred_actors Sally Field
kb: Forrest Gump directed_by Robert Zemeckis
T : Which movie did Tom Hanks star in ?
S : I don't know. What's the answer?
T : The answer is Forrest Gump.
T/S : Conversation History.
T : Which movie did Tom Hanks star in ?
S : Forrest Gump
T : That's correct. (+)

**Task 6: Missing Answer Entity (AQ)**
kb: Larry Crowne directed_by Tom Hanks
~~kb: Forrest Gump starred_actors Tom Hanks~~
~~kb: Forrest Gump starred_actors Sally Field~~
~~kb: Forrest Gump directed_by Robert Zemeckis~~
T : Which movie did Tom Hanks star in ?
S : I don't know. What's the answer?
T : The answer is Forrest Gump.
T/S : Conversation History.
T : Which movie did Tom Hanks star in ?
S : Forrest Gump
T : That's correct. (+)

**Task 7: Missing Relation Entity (AQ)**
kb: Larry Crowne directed_by Tom Hanks
~~kb: Forrest Gump starred_actors Tom Hanks,~~
~~kb: Forrest Gump starred_actors Sally Field~~
kb: Forrest Gump directed_by Robert Zemeckis
T : Which movie did Tom Hanks star in ?
S : I don't know. What's the answer?
T : The answer is Forrest Gump.
T/S : Conversation History.
T : Which movie did Tom Hanks star in ?
S : Forrest Gump
T : That's correct. (+)

.

**Task 8: Missing Triple (AQ)**
kb: Larry Crowne directed_by Tom Hanks
~~kb: Forrest Gump starred_actors Tom Hanks~~
kb: Forrest Gump starred_actors Sally Field
kb: Forrest Gump directed_by Robert Zemeckis
T : Which movie did Tom Hanks star in ?
S : I don't know. What's the answer?
T : The answer is Forrest Gump.
T/S : Conversation History.
T : Which movie did Tom Hanks star in ?
S : Forrest Gump
T : That's correct. (+)

**Task 9: Missing Everything (AQ)**
~~kb: Larry Crowne directed_by Tom Hanks~~
~~kb: Forrest Gump starred_actors Tom Hanks,~~
~~kb: Forrest Gump starred_actors Sally Field~~
~~kb: Forrest Gump directed_by Robert Zemeckis~~
T : Which movie did Tom Hanks star in ?
S : I don't know. What's the answer?
T : The answer is Forrest Gump.
T/S : Conversation History.
T : Which movie did Tom Hanks star in ?
S : Forrest Gump
T : That's correct. (+)

**Missing Answer Entity**: The answer entity to the question is unknown to the bot. All KB facts that contain the answer entity will be hidden. Hence, in *Task 6* of Figure 3, all KB facts containing the answer entity Forrest Gump will be hidden from the bot.

**Missing Relation Entity**: The relation type is unknown to the bot. In *Task 7* of Figure 3, all KB facts that express the relation starred_actors are hidden from the bot.

**Missing Triples**: The triple that expresses the relation between the question entity and the answer entity is hidden from the bot. In *Task 8* of Figure 3, the triple "Forrest Gump (question entity) starred_actors Tom Hanks (answer entity)" will be hidden.

**Missing Everything**: The question entity, the relation entity, the answer entity are all missing from the KB. All KB facts in *Task 9* of Figure 3 will be removed since they either contain the relation entity (i.e., starred_actors), the question entity (i.e., Forrest Gump) or the answer entity Tom Hanks.

## 4 TRAIN/TEST REGIME

We now discuss in detail the regimes we used to train and test our models, which are divided between evaluation within our simulator and using real data collected via Mechanical Turk.

### 4.1 SIMULATOR

Using our simulator, our objective was twofold. We first wanted to validate the usefulness of asking questions in all the settings described in Section 3. Second, we wanted to assess the ability of our student bot to learn *when* to ask questions. In order to accomplish these two objectives we explored training our models with our simulator using two methodologies, namely, Offline Supervised Learning and Online Reinforcement Learning.

#### 4.1.1 OFFLINE SUPERVISED LEARNING

The motivation behind training our student models in an offline supervised setting was primarily to test the usefulness of the ability to ask questions. The dialogues are generated as described in the previous section, and the bot's role is generated with a fixed policy. We chose a policy where answers to the teacher's questions are correct answers 50% of the time, and incorrect otherwise, to add a degree of realism. Similarly, in tasks where questions can be irrelevant they are only asked correctly 50% of the time.[6]

The offline setting explores different combinations of training and testing scenarios, which mimic different situations in the real world. The aim is to understand when and how observing interactions between two agents can help the bot improve its performance for different tasks. As a result we construct training and test sets in three ways across all tasks, resulting in 9 different scenarios per task, each of which correspond to a real world scenario.

The three training sets we generated are referred to as **TrainQA, TrainAQ, and TrainMix**. TrainQA follows the QA setting discussed in the previous section: the bot never asks questions and only tries to immediately answer. TrainAQ follows the AQ setting: the student, before answering, first always asks a question in response to the teacher's original question. TrainMix is a combination of the two where 50% of time the student asks a question and 50% does not.

The three test sets we generated are referred to as **TestQA, TestAQ, and TestModelAQ**. TestQA and TestAQ are generated similarly to TrainQA and TrainAQ, but using a perfect fixed policy (rather than 50% correct) for evaluation purposes. In the TestModelAQ setting the model has to get the form of the question correct as well. In the *Question Verification* and *Knowledge Verification* tasks there are many possible ways of forming the question and some of them are correct – the model has to choose the right question to ask. E.g. it should ask "Does it have something to do with the fact that Larry Crowne directed by Tom Hanks?"rather than "Does it have something to do with the fact that Forrest Gump directed by Robert Zemeckis?" when the latter is irrelevant (the candidate list of questions is generated from the known knowledge base entries with respect to that question). The policy is trained using either the TrainAQ or TrainMix set, depending on the training scenario. The teacher will reply to the question, giving positive feedback if the student's question is correct and no response and negative feedback otherwise. The student will then give the final answer. The difference between TestModelAQ and TestAQ only exists in the *Question Verification* and *Knowledge Verification* tasks; in other tasks there is only one way to ask the question and TestModelAQ and TestAQ are identical.

To summarize, for every task listed in Section 3 we train one model for each of the three training sets (TrainQA, TrainAQ, TrainMix) and test each of these models on the three test sets (TestQA, TestAQ, and TestModelAQ), resulting in 9 combinations. For the purpose of notation the train/test combination is denoted by "TrainSetting+TestSetting". For example, *TrainAQ+TestQA* denotes a model which is trained using the TrainAQ dataset and tested on TestQA dataset. Each combination has a real world interpretation. For instance, *TrainAQ+TestQA* would refer to a scenario where a student can ask the teacher questions during learning but cannot to do so while taking an exam. Similarly, *TrainQA+TestQA* describes a stoic teacher that never answers a student's question at either learning or examination time. The setting *TrainQA+TestAQ* corresponds to the case where a lazy

---

[6]This only makes sense in tasks like Question or Knowledge Verification. In tasks where the question is static such as 'What do you mean?" there is no way to ask an irrelevant question, and we do not use this policy.

student never asks question at learning time but gets anxious during the examination and always asks a question.

### 4.1.2 ONLINE REINFORCEMENT LEARNING (RL)

We also explored scenarios where the student learns the ability to decide when to ask a question. In other words, the student learns how to learn.

Although it is in the interest of the student to ask questions at every step of the conversation, since the response to its question will contain extra information, we don't want our model to learn this behavior. Each time a human student asks a question, there's a cost associated with that action. This cost is a reflection of the patience of the teacher, or more generally of the users interacting with the bot in the wild: users won't find the bot engaging if it always asks clarification questions. The student should thus be judicious about asking questions and learn when and what to ask. For instance, if the student is confident about the answer, there is no need for it to ask. Or, if the teacher's question is so hard that clarification is unlikely to help enough to get the answer right, then it should also refrain from asking.

We now discuss how we model this problem under the Reinforcement Learning framework. The bot is presented with KB facts (some facts might be missing depending on the task) and a question. It needs to decide whether to ask a question or not at this point. The decision whether to ask is made by a binary policy $P_{RLQuestion}$. If the student chooses to ask a question, it will be penalized by $cost_{AQ}$. We explored different values of $cost_{AQ}$ ranging from $[0, 2]$, which we consider as modeling the patience of the teacher. The goal of this setting is to find the best policy for asking/not-asking questions which would lead to the highest cumulative reward. The teacher will appropriately reply if the student asks a question. The student will eventually give an answer to the teacher's initial question at the end using the policy $P_{RLAnswer}$, regardless of whether it had asked a question. The student will get a reward of $+1$ if its final answer is correct and $-1$ otherwise. Note that the student can ask at most one question and that the type of question is always specified by the task under consideration. The final reward the student gets is the cumulative reward over the current dialogue episode. In particular the reward structure we propose is the following:

|  | Asking Question | Not asking Question |
|---|---|---|
| Final Answer Correct | 1-$cost_{AQ}$ | 1 |
| Final Answer Incorrect | -1-$cost_{AQ}$ | -1 |

Table 1: Reward structure for the Reinforcement Learning setting.

For each of the tasks described in Section 3, we consider three different RL scenarios.
**Good-Student**: The student will be presented with all relevant KB facts. There are no misspellings or unknown words in the teacher's question. This represents a knowledgable student in the real world that knows as much as it needs to know (e.g., a large knowledge base, large vocabulary). This setting is identical across all missing entity tasks (5 - 9).
**Poor-Student**: The KB facts or the questions presented to the student are flawed depending on each task. For example, for the *Question Clarification* tasks, the student does not understand the question due to spelling mistakes. For the *Missing Question Entity* task the entity that the teacher asks about is unknown by the student and all facts containing the entity will be hidden from the student. This setting is similar to a student that is underprepared for the tasks.
**Medium-Student**: The combination of the previous two settings where for $50\%$ of the questions, the student has access to the full KB and there are no new words or phrases or entities in the question, and $50\%$ of the time the question and KB are taken from the *Poor-Student* setting.

### 4.2 MECHANICAL TURK DATA

Finally, to validate our approach beyond our simulator by using real language, we collected data via Amazon Mechanical Turk. Due to the cost of data collection, we focused on real language versions of Tasks 4 (Knowledge Verification) and 8 (Missing Triple), see Secs. 3.2 and 3.3 for the simulator versions. That is, we collect dialogues and use them in an offline supervised learning setup similar to Section 4.1.1. This setup allows easily reproducible experiments.

For Mechanical Turk Task 4, the bot is asked a question by a human teacher, but before answering can ask the human if the question is related to one of the facts it knows about from its memory.

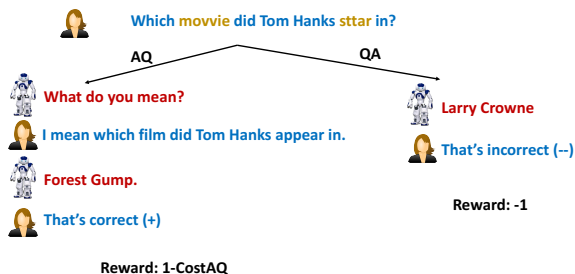

Figure 4: An illustration of the *poor-student* setting for RL Task 1 (Question Paraphrase).

It is then required to answer the original question, after some additional dialog turns relating to other question/answer pairs (called "conversational history", as before). For Task 8, the bot is asked a question by a human but lacks the triple in its memory that would be needed to answer it. It is allowed to ask for the missing information, the human responds to the question in free-form language. The bot is then required to answer the original question, again after some "conversational history" has transpired.

We collect around 10,000 episodes (dialogues) for training, 1000 for validation, and 2500 for testing for each of the two tasks. In each case, we give instructions to the Turkers that still follow the original form of the task, but make the tasks contain realistic language written by humans. The instructions given to the Turkers are given in the appendix.

For both tasks, while the human turkers replace the simulator that the bot was previously conversing with, the bot's dialogue actions (capabilities) are essentially unchanged from before. That is, when answering questions, now the bot is required to answer a human's questions rather than templated questions from the simulator. When the bot is asking questions, the bot still asks in the same form as before, e.g. questions like "Does it have something to do with X" for Task 4 or "I don't know. What's the answer?" for Task 8. However, now its questions are answered by a human. In both cases (asking and answering) the human data is richer with potentially more complex language and lexical variability. Examples of the collected dialogues are given in Figure 5.

---

**MTurk Task 4: Knowledge Verification (AQ):**
  kb: In Their Skin written_by Joshua Close, Jeremy Power Regimbal
  kb: In Their Skin starred_actors Selma Blair, James D'arcy, Rachel Miner, Joshua Close
  T: What film was written by Joshua Close?
  S: Does it have something to do with "In Their Skin starred_actors Joshua Close"?
  T: No we were looking for who wrote the film not who starred in it.
  . . .
**MTurk Task 8: Missing Triple (AQ):**
  T :What movies does Timothy Scott Bogart have writing credit for?
  S: I don't know. What's the answer?
  T: Extreme Ops or The I Inside is the right answer.
  . . .

---

Figure 5: Sample dialogues for Mechanical Turk versions of Tasks 4 and 8. Compared to the original tasks (see Figs 2 and 3) the teacher's questions, and the teacher responses to the student's questions, are written by humans and are more complex and contain more variety.

## 5 MODELS

For both offline supervised and online RL settings, we use the *End-to-End Memory Network* model (MemN2N) (Sukhbaatar et al., 2015) as a backbone. The model takes as input the last utterance of the dialogue history (the question from the teacher) as well as a set of memory contexts including short-term memories (the dialogue history between the bot and the teacher) and long-term memories

(the knowledge base facts that the bot has access to), and outputs a label. We refer readers to the Appendix for more details about MemN2N.

**Offline Supervised Settings**: The first learning strategy we adopt is the reward-based imitation strategy (denoted *vanilla-MemN2N*) described in (Weston, 2016), where at training time, the model maximizes the log likelihood probability of the correct answers the student gave (examples with incorrect final answers are discarded). Candidate answers are words that appear in the memories, which means the bot can only predict the entities that it has seen or known before.

We also use a variation of MemN2N called "context MemN2N" (*Cont-MemN2N* for short) where we replace each word's embedding with the average of its embedding (random for unseen words) and the embeddings of the other words that appear around it. We use both the preceeding and following words as context and the number of context words is a hyperparameter selected on the dev set.

An issue with both *vanilla-MemN2N* and *Cont-MemN2N* is that the model only makes use of the bot's answers as signals and ignores the teacher's feedback. We thus propose to use a model that jointly predicts the bot's answers and the teacher's feedback (denoted as *TrainQA (+FP)*). The bot's answers are predicted using a *vanilla-MemN2N* and the teacher's feedback is predicted using the *Forward Prediction* (FP) model as described in (Weston, 2016). We refer the readers to the Appendix for the FP model details. At training time, the models learn to jointly predict the teacher's feedback and the answers with positive reward. At test time, the model will only predict the bot's answer.

For the *TestModelAQ* setting described in Section 4, the model needs to decide the question to ask. Again, we use vanilla-MemN2N that takes as input the question and contexts, and outputs the question the bot will ask.

**Online RL Settings**: A binary vanilla-MemN2N (denoted as $P_{RL}(Question)$) is used to decide whether the bot should or should not ask a question, with the teacher replying if the bot does ask something. A second MemN2N is then used to decide the bot's answer, denoted as $P_{RL}(Answer)$. $P_{RL}(Answer)$ for $QA$ and $AQ$ are two separate models, which means the bot will use different models for final-answer prediction depending on whether it chooses to ask a question or not.[7]

We use the REINFORCE algorithm (Williams, 1992) to update $P_{RL}(Question)$ and $P_{RL}(Answer)$. For each dialogue, the bot takes two sequential actions $(a_1, a_2)$: to ask or not to ask a question (denoted as $a_1$); and guessing the final answer (denoted as $a_2$). Let $r(a_1, a_2)$ denote the cumulative reward for the dialogue episode, computed using Table 1. The gradient to update the policy is given by:

$$p(a_1, a_2) = P_{RL}(Question)(a_1) \cdot P_{RL}(answer)(a_2)$$
$$\nabla J(\theta) \approx \nabla \log p(a_1, a_2)[r(a_1, a_2) - b] \tag{1}$$

where $b$ is the baseline value, which is estimated using another MemN2N model that takes as input the query $x$ and memory $C$, and outputs a scalar $b$ denoting the estimation of the future reward. The baseline model is trained by minimizing the mean squared loss between the estimated reward $b$ and actual cumulative reward $r$, $||r - b||^2$. We refer the readers to (Ranzato et al., 2015; Zaremba & Sutskever, 2015) for more details. The baseline estimator model is independent from the policy models and the error is not backpropagated back to them.

In practice, we find the following training strategy yields better results: first train only $P_{RL}(answer)$, updating gradients only for the policy that predicts the final answer. After the bot's final-answer policy is sufficiently learned, train both policies in parallel[8]. This has a real-world analogy where the bot first learns the basics of the task, and then learns to improve its performance via a question-asking policy tailored to the user's patience (represented by $cost_{AQ}$) and its own ability to asnwer questions.

---

[7]An alternative is to train one single model for final answer prediction in both *AQ* and *QA* cases, similar to the *TrainMix* setting in the supervised learning setting. But we find training *AQ* and *QA* separately for the final answer prediction yields a little better result than the single model setting.

[8] We implement this by running 16 epochs in total, updating only the model's policy for final answers in the first 8 epochs while updating both policies during the second 8 epochs. We pick the model that achieves the best reward on the dev set during the final 8 epochs. Due to relatively large variance for RL models, we repeat each task 5 times and keep the best model on each task.

| | Question Clarification | | | | Knowledge Operation | | | |
|---|---|---|---|---|---|---|---|---|
| | Task 1: Q. Paraphrase | | Task 2: Q. Verification | | Task 3: Ask For Relevant K. | | Task 4: K. Verification | |
| Train \Test | TestQA | TestAQ | TestQA | TestAQ | TestQA | TestAQ | TestQA | TestAQ |
| TrainQA (Context) | 0.754 | 0.726 | 0.742 | 0.684 | 0.883 | 0.947 | 0.888 | 0.959 |
| TrainAQ (Context) | 0.640 | 0.889 | 0.643 | 0.807 | 0.716 | 0.985 | 0.852 | 0.987 |
| TrainMix (Context) | 0.751 | 0.846 | 0.740 | 0.789 | 0.870 | 0.985 | 0.875 | 0.985 |

| | Knowledge Acquisition | | | | | | | | | |
|---|---|---|---|---|---|---|---|---|---|---|
| Train \Test | TestQA | TestAQ | TestQA | TestAQ | TestQA | TestAQ | TestQA | TestAQ | TestQA | TestAQ |
| | Task 5: Q. Entity | | Task 6: Answer Entity | | Task 7: Relation Entity | | Task 8: Triple | | Task 9: Everything | |
| TrainQA (Context) | <0.01 | 0.224 | <0.01 | 0.120 | 0.241 | 0.301 | 0.339 | 0.251 | <0.01 | 0.058 |
| TrainAQ (Context) | <0.01 | 0.639 | <0.01 | 0.885 | 0.143 | 0.893 | 0.154 | 0.884 | <0.01 | 0.908 |
| TrainMix (Context) | <0.01 | 0.632 | <0.01 | 0.852 | 0.216 | 0.898 | 0.298 | 0.886 | <0.01 | 0.903 |

Table 2: Results for Cont-MemN2N on different tasks.

# 6 EXPERIMENTS

## 6.1 SIMULATOR

**Offline Results**: Offline results are presented in Tables 2, 7 and 8 (the latter two are in the appendix). Table 7 presents results for the *vanilla-MemN2N* and *Forward Prediction* models. Table 2 presents results for *Cont-MemN2N*, which is better at handling unknown words. We repeat each experiment 10 times and report the best result. Finally, Table 8 presents results for the test scenario where the bot itself chooses when to ask questions. Observations can be summarized as as follows:

- Asking questions helps at test time, which is intuitive since it provides additional evidence:

- *TrainAQ+TestAQ* (questions can be asked at both training and test time) performs the best across all the settings.

- *TrainQA+TestAQ* (questions can be asked at training time but not at test time) performs worse than *TrainQA+TestQA* (questions can be asked at neither training nor test time) in tasks *Question Clarification* and *Knowledge Operation* due to the discrepancy between training and testing.

- *TrainQA+TestAQ* performs better than *TrainQA+TestQA* on all *Knowledge Acquisition* tasks, the only exception being the *Cont-MemN2N* model on the *Missing Triple* setting. The explanation is that for most tasks in *Knowledge Acquisition*, the learner has no chance of giving the correct answer without asking questions. The benefit from asking is thus large enough to compensate for the negative effect introduced by data discrepancy between training and test time.

- *TrainMix* offers flexibility in bridging the gap between datasets generated using QA and AQ, very slightly underperforming *TrainAQ+TestAQ*, but gives competitive results on both *TestQA* and *TestAQ* in the *Question Clarification* and *Knowledge Operations* tasks.

- *TrainAQ+TestQA* (allowing questions at training time but forbid questions at test time) performs the worst, even worse than *TrainQA+TestQA*. This has a real-world analogy where a student becomes dependent on the teacher answering their questions, later struggling to answer the test questions without help.

- In the *Missing Question Entity* task (the student does not know about the question entity), the *Missing Answer Entity* task (the student does not know about the answer entity), and *Missing Everything* task, the bot achieves accuracy less than 0.01 if not asking questions at test time (i.e., *TestQA*).

- The performance of *TestModelAQ*, where the bot relies on its model to ask questions at test time (and thus can ask irrelevant questions) performs similarly to asking the correct question at test time (*TestAQ*) and better than not asking questions (*TestQA*).

- *Cont-MemN2N* significantly outperforms *vanilla-MemN2N*. One explanation is that considering context provides significant evidence distinguishing correct answers from candidates in the dialogue history, especially in cases where the model encounters unfamiliar words.

**RL Results** For the RL settings, we present results for Task 2 (*Question Verification*) and Task 6 (*Missing Answer Entities*) in Figure 6. Task 2 represents scenarios where different types of student

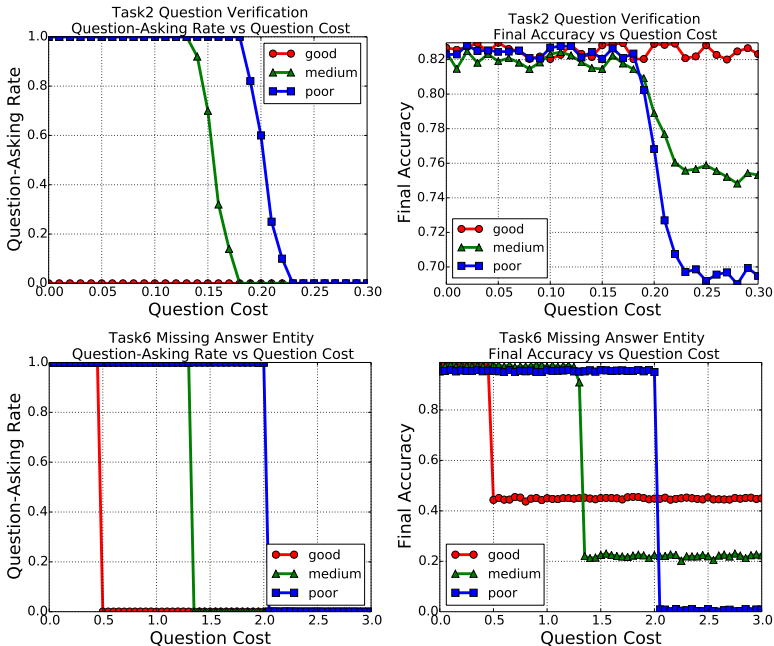

Figure 6: Results of online learning for Task 2 and Task 6

have different abilities to correctly answer questions (e.g., a poor student can still sometimes give correct answers even when they do not fully understand the question). Task 6 represents tasks where a poor learner who lacks the knowledge necessary to answer the question can hardly give a correct answer. All types of students including the good student will theoretically benefit from asking questions (asking for the correct answer) in Task 6. We show the percentage of question-asking versus the cost of AQ on the test set and the accuracy of question-answering on the test set vs the cost of AQ. Our main findings were:

- A good student does not need to ask questions in Task 2 (*Question Verification*), because they already understand the question. The student will raise questions asking for the correct answer when cost is low for Task 6 (*Missing Answer Entities*).

- A poor student always asks questions when the cost is low. As the cost increases, the frequency of question-asking declines.

- As the AQ cost increases gradually, good students will stop asking questions earlier than the medium and poor students. The explanation is intuitive: poor students benefit more from asking questions than good students, so they continue asking even with higher penalties.

- As the probability of question-asking declines, the accuracy for poor and medium students drops. Good students are more resilient to not asking questions.

## 6.2 MECHANICAL TURK

Results for the Mechanical Turk Tasks are given in Table 3. We again compare vanilla-MemN2N and Cont-MemN2N, using the same TrainAQ/TrainQA and TestAQ/TestQA combinations as before, for Tasks 4 and 8 as described in Section 4.2. We tune hyperparameters on the validation set and repeat each experiment 10 times and report the best result.

While performance is lower than on the related Task 4 and Task 8 simulator tasks, we still arrive at the same trends and conclusions when real data from humans is used. The performance was expected to be lower because (i) real data has more lexical variety, complexity and noise; and (ii) the training set was smaller due to data collection costs (10k vs. 180k). We perform an analysis of the difference between simulated and real training data (or combining the two) in the appendix, which shows that using real data is indeed important and measurably superior to using simulated data.

| | vanilla-MemN2N | | | | Cont-MemN2N | | | |
| | Task 4: K. Verification | | Task 8: Triple | | Task 4: K. Verification | | Task 8: Triple | |
| Train \ Test | TestQA | TestAQ | TestQA | TestAQ | TestQA | TestAQ | TestQA | TestAQ |
| TrainQA | 0.331 | 0.313 | 0.133 | 0.162 | 0.712 | 0.703 | 0.308 | 0.234 |
| TrainAQ | 0.318 | 0.375 | 0.072 | 0.422 | 0.679 | 0.774 | 0.137 | 0.797 |

Table 3: Mechanical Turk Task Results. Asking Questions (AQ) outperforms only answering questions without asking (QA).

More importantly, the same main conclusion is observed as before: TrainAQ+TestAQ (questions can be asked at both training and test time) performs the best across all the settings. That is, we show that a bot asking questions to humans learns to outperform one that only answers them.

## 7  CONCLUSIONS

In this paper, we explored how an intelligent agent can benefit from interacting with users by asking questions. We developed tasks where interaction via asking questions is desired. We explore both online and offline settings that mimic different real world situations and show that in most cases, teaching a bot to interact with humans facilitates language understanding, and consequently leads to better question answering ability.

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

## Appendix

**End-to-End Memory Networks** The input to an end-to-end memory network model (MemN2N) is the last utterance of the dialogue history $x$ as well as a set of memories (context) ($C=c_1$, $c_2$, ..., $c_N$). Memory $C$ encodes both short-term memory, e..g, dialogue histories between the bot and the teacher and long-term memories, e.g., the knowledgebase facts that the bot has access to. Given the input $x$ and $C$, the goal is to produce an output/label $a$.

In the first step, the query $x$ is transformed to a vector representation $u_0$ by summing up its constituent word embeddings: $u_0 = Ax$. The input x is a bag-of-words vector and $A$ is the $d \times V$ word embedding matrix where $d$ denotes the vector dimensionality and $V$ denotes the vocabulary size. Each memory $c_i$ is similarly transformed to vector $m_i$. The model will read information from the memory by linking input representation $q$ with memory vectors $m_i$ using softmax weights:

$$o_1 = \sum_i p_i^1 m_i \qquad p_i^1 = \texttt{softmax}(u_0^T m_i) \qquad (2)$$

The goal is to select memories relevant to the last utterance $x$, i.e., the memories with large values of $p_i^1$. The queried memory vector $o_1$ is the weighted sum of memory vectors. The queried memory vector $o_1$ will be added on top of original input, $u_1 = o_1 + u_0$. $u_1$ is then used to query the memory vector. Such a process is repeated by querying the memory N times (so called "hops"). $N$ is set to three in all experiments in this paper.

In the end, $u_N$ is input to a softmax function for the final prediction:

$$a = \texttt{softmax}(u_N^T y_1, u_N^T y_2, ..., u_N^T y_L) \qquad (3)$$

where $L$ denotes the number of candidate answers and $y$ denotes the representation of the answer. If the answer is a word, $y$ is the corresponding word embedding. If the answer is a sentence, $y$ is the embedding for the sentence achieved in the same way as we obtain embeddings for query $x$ and memory $c$.

**Reward Based Imitation (RBI) and Forward Prediction (FP)** *RBI* and *FP* are two dialogue learning strategies proposed in (Weston, 2016) by harnessing different types of dialogue signals. *RBI* handles the case where the reward or the correctness of a bot's answer is explicitly given (for example, +1 if the bot's answer is correct and 0 otherwise). The model is directly trained to predict the correct answers (with label 1) at training time, which can be done using End-to-End Memory Networks (*MemN2N*) (Sukhbaatar et al., 2015) that map a dialogue input to a prediction.

*FP* handles the situation where a real-valued reward for a bot's answer is not available, meaning that there is no +1 or 0 labels paired with a student's utterance. However, the teacher will give a response to the bot's answer, taking the form of a dialogue utterance. More formally, suppose that $x$ denotes the teacher's question and $C = c_1, c_2, ..., c_N$ denotes the dialogue history. In our *AQ* settings, the bot will ask a question $a$ regarding the teacher's question, denoted as $a \in \mathbb{A}$, where $\mathbb{A}$ denotes the student's question pool. The teacher will provide an utterance in response to the student question $a$. In *FP*, the model first maps the teacher's initial question $x$ and dialogue history $C$ to vector representation $u$ using a memory network with multiple hops. Then the model will perform another hopof attention over all possible student's questions in $\mathbb{A}$, with an additional part that incorporates the information of which candidate (i.e., $a$) was actually selected in the dialogue:

$$p_{\hat{a}} = \texttt{softmax}(u^T y_{\hat{a}}) \quad o = \sum_{\hat{a} \in \mathbb{A}} p_{\hat{a}}(y_{\hat{a}} + \beta \cdot \mathbf{1}[\hat{a} = a]) \tag{4}$$

where $y_{\hat{a}}$ denotes the vector representation for the student's question candidate $\hat{a}$. $\beta$ is a d-dimensional vector to signify the actual action $a$ that the student chooses. For tasks where the student only has one way to ask questions (e.g., "what do you mean"), there is no need to perform hops of attention over candidates since the cardinality of $\mathbb{A}$ is just 1. We thus directly assign a probability of 1 to the student's question, making $o$ the sum of vector representation of $y_a$ and $\beta$.

$o$ is then combined with $u$ to predict the teacher's feedback $t$ using a softmax:

$$u_1 = o + u \quad t = \texttt{softmax}(u_1^T x_{r_1}, u_1^T x_{r_2}, ..., u_1^T x_{r_N}) \tag{5}$$

where $x_{r_i}$ denotes the embedding for the $i^{th}$ response.

**Dialogue Simulator** In this section we further detail the simulator and the datasets we generated in order to realize the various scenarios discussed in Section 3. We focused on the problem of movieQA where we adapted the WikiMovies dataset proposed in Weston et al. (2015). The dataset consists of roughly 100k questions with over 75k entities from the open movie dataset (OMDb).

Each dialogue generated by the simulator takes place between a student and a teacher. The simulator samples a random question from the WikiMovies dataset and fetches the set of all KB facts relevant to the chosen question. This question is assumed to be the one the teacher asks its student, and is referred to as the "original" question. The student is first presented with the relevant KB facts followed by the original question. Providing the KB facts to the student allows us to control the exact knowledge the student is given access to while answering the questions. At this point, depending on the task at hand and the student's ability to answer, the student might choose to directly answer it or ask a "followup" question. The nature of the followup question will depend on the scenario under consideration. If the student answers the question, it gets a response from the teacher about its correctness and the conversation ends. However if the student poses a followup question, the teacher gives an appropriate response, which should give additional information to the student to answer the original question. In order to make things more complicated, the simulator pads the conversation with several unrelated student-teacher question-answer pairs. These question-answer pairs can be viewed as distractions and are used to test the student's ability to remember the additional knowledge provided by the teacher after it was queried. For each dialogue, the simulator incorporates 5 such pairs (10 sentences). We refer to these pairs as *conversational histories*.

For the *QA* setting (see Section 3), the dialogues generated by the simulator are such that the student never asks a clarification question. Instead, it simply responds to the original question, even if it is wrong. For the dialogs in the *AQ* setting, the student *always* asks a clarification question. The nature of the question asked is dependent on the scenario (whether it is *Question Clarification*, *Knowledge Operation*, or *Knowledge Acquisition*) under consideration. In order to simulate the case where the student sometimes choses to directly answer the original question and at other times choses to ask question, we created training datasets, which were a combination of *QA* and *AQ* (called "Mixed"). For all these cases, the student needs to give an answer to the teacher's original question at the end.

**Instructions given to Turkers**

These are the instructions given for the textual feedback Mechanical Turk task (we also constructed a separate task to collect the questions to ask the bot with similar instructions, not described here):

*Task 4 (answers to bot's questions):*

Title: Write brief responses to given dialogue exchanges (about 15 min)

Description: Write a brief response answering a provided question (25 questions per HIT).

Directions:

Each task consists of the following triplets:
1) a question by the teacher
2) the correct answer(s) to the question (separated by "OR"), unknown to the student
3) a clarifying question asking for feedback from the teacher
Consider the scenario where you are the teacher and have already asked the question, and received the reply from the student. Please compose a brief response replying to the student's question. The correct answers are provided so that you know whether the student's question was relevant or not.
For example, given 1) question: "what is a color in the united states flag?"; 2) correct answer: "white OR blue OR red"; 3) student reply: "does this have to do with 'US Flag has_colors red,white,blue'?", your response could be something like "that's right!"; for 3) reply: "does this have to do with 'United States has_population 320 million", you might say "No, that fact is not relevant" or "Not really".
Please vary responses and try to minimize spelling mistakes. If the same responses are copied/pasted or similar responses are overused, we'll reject the HIT. Avoid naming the student or addressing "the class" directly. We will consider bonuses for higher quality responses during review.

*Task 8: answers to bot's questions:*

Title: Write brief responses to given dialogue exchanges (about 10 min)

Description: Write a sentence describing the answer to a question (25 questions per HIT).

Directions:

Each task consists of the following triplets:
1) a question by the teacher
2) the correct answer(s) to the question (separated by "OR"), unknown to the student
3) a question from the student asking the teacher for the answer
Consider the scenario where you are the teacher and have already asked the question, and received the reply from the student. Please compose a brief response replying to the student's question. The correct answers are provided so that you know which answers to provide.
For example, given 1) question: "what is a color in the united states flag?"; 2) correct answer: "white OR blue OR red"; 3) student reply: "i dont know. what's the answer ?", your response could be something like "the color white is in the US flag" or "blue and red both appear in it".
Please vary responses and try to minimize spelling mistakes, and do not include the capitalized "OR" in your response. If the same responses are copied/pasted or similar responses are overused, we'll reject the HIT. You don't need to mention every correct answer in your response. Avoid naming the student or addressing "the class" directly. We will consider bonuses for higher quality responses during review.

**Additional Mechanical Turk Experiments**

Here we provide additional experiments to supplement the ones described in Section 6.2. In the main paper, results were shown when training and testing on the collected Mechanical Turk data (around 10,000 episodes of training dialogues for training). As we collected the data in the same settings as Task 4 and 8 of our simulator, we could also consider supplementing training with simulated data as well, of which we have a larger amount (over 100,000 episodes). Note this is *only for training*, we will still test on the real (Mechanical Turk collected) data. Although the simulated data has less lexical variety as it is built from templates, the larger size might obtain improve results.

Results are given Table 5 when training on the combination of real and simulator data, and testing on real data. This should be compared to training on only the real data (Table 4) and only on the simulator data (Table 6). The best results are obtained from the combination of simulator and real data. The best real data only results (selecting over algorithm and training strategy) on both tasks outperform the best results using simulator data, i.e. using Cont-MemN2N with the Train AQ / TestAQ setting) 0.774 and 0.797 is obtained vs. 0.714 and 0.788 for Tasks 4 and 8 respectively. This

is despite there being far fewer examples of real data compared to simulator data. Overall we obtain two main conclusions from this additional experiment: (i) real data is indeed measurably superior to simulated data for training our models; (ii) in all cases (across different algorithms, tasks and data types – be they real data, simulated data or combinations) the bot asking questions (AQ) outperforms it only answering questions and not asking them (QA). The latter reinforces the main result of the paper.

| Train \ Test | vanilla-MemN2N | | | | Cont-MemN2N | | | |
|---|---|---|---|---|---|---|---|---|
| | Task 4: K. Verification | | Task 8: Triple | | Task 4: K. Verification | | Task 8: Triple | |
| | TestQA | TestAQ | TestQA | TestAQ | TestQA | TestAQ | TestQA | TestAQ |
| TrainQA | 0.331 | 0.313 | 0.133 | 0.162 | 0.712 | 0.703 | 0.308 | 0.234 |
| TrainAQ | 0.318 | 0.375 | 0.072 | 0.422 | 0.679 | 0.774 | 0.137 | 0.797 |

Table 4: Mechanical Turk Task Results, using real data for training and testing.

| Train \ Test | vanilla-MemN2N | | | | Cont-MemN2N | | | |
|---|---|---|---|---|---|---|---|---|
| | Task 4: K. Verification | | Task 8: Triple | | Task 4: K. Verification | | Task 8: Triple | |
| | TestQA | TestAQ | TestQA | TestAQ | TestQA | TestAQ | TestQA | TestAQ |
| TrainQA | 0.356 | 0.311 | 0.128 | 0.174 | 0.733 | 0.717 | 0.368 | 0.352 |
| TrainAQ | 0.340 | 0.445 | 0.150 | 0.487 | 0.704 | 0.792 | 0.251 | 0.825 |

Table 5: Results on Mechanical Turk Tasks using a combination of real and simulated data for training, testing on real data.

| Train \ Test | vanilla-MemN2N | | | | Cont-MemN2N | | | |
|---|---|---|---|---|---|---|---|---|
| | Task 4: K. Verification | | Task 8: Triple | | Task 4: K. Verification | | Task 8: Triple | |
| | TestQA | TestAQ | TestQA | TestAQ | TestQA | TestAQ | TestQA | TestAQ |
| TrainQA | 0.340 | 0.311 | 0.120 | 0.165 | 0.665 | 0.648 | 0.349 | 0.342 |
| TrainAQ | 0.326 | 0.390 | 0.067 | 0.405 | 0.642 | 0.714 | 0.197 | 0.788 |

Table 6: Results on Mechanical Turk Tasks using only simulated data for training, but testing on real data.

**Additional Offline Supervised Learning Experiments**

| Train \ Test | Question Clarification | | | | Knowledge Operation | | | |
|---|---|---|---|---|---|---|---|---|
| | Task 1: Q. Paraphrase | | Task 2: Q. Verification | | Task 3: Ask For Relevant K. | | Task 4: K. Verification | |
| | TestQA | TestAQ | TestQA | TestAQ | TestQA | TestAQ | TestQA | TestAQ |
| TrainQA | 0.338 | 0.284 | 0.340 | 0.271 | 0.462 | 0.344 | 0.482 | 0.322 |
| TrainAQ | 0.213 | 0.450 | 0.225 | 0.373 | 0.187 | 0.632 | 0.283 | 0.540 |
| TrainAQ(+FP) | 0.288 | 0.464 | 0.146 | 0.320 | 0.342 | 0.631 | 0.311 | 0.524 |
| TrainMix | 0.326 | 0.373 | 0.329 | 0.326 | 0.442 | 0.558 | 0.476 | 0.491 |

| Train \ Test | Knowledge Acquisition | | | | | | | | | |
|---|---|---|---|---|---|---|---|---|---|---|
| | TestQA | TestAQ | TestQA | TestAQ | TestQA | TestAQ | TestQA | TestAQ | TestQA | TestAQ |
| | Task 5: Q. Entity | | Task 6: Answer Entity | | Task 7: Relation Entity | | Task 8: Triple | | Task 9: Everything | |
| TrainQA (vanila) | < 0.01 | 0.223 | <0.01 | <0.01 | 0.109 | 0.129 | 0.201 | 0.259 | <0.01 | <0.01 |
| TrainAQ (vanila) | < 0.01 | 0.660 | <0.01 | <0.01 | 0.082 | 0.156 | 0.124 | 0.664 | <0.01 | <0.01 |
| TrainAQ(+FP) | < 0.01 | 0.742 | < 0.01 | < 0.01 | 0.085 | 0.188 | 0.064 | 0.702 | <0.01 | <0.01 |
| Mix (vanila) | <0.01 | 0.630 | <0.01 | <0.01 | 0.070 | 0.152 | 0.180 | 0.572 | <0.01 | <0.01 |

Table 7: Results for offline settings using memory networks.

| | Question Clarification | Knowledge Acquisition |
|---|---|---|
| | Task 2: Q. Verification | Task 4: K. Verification |
| | TestModelAQ | TestModelAQ |
| TrainAQ | 0.382 | 0.480 |
| TrainAQ(+FP) | 0.344 | 0.501 |
| TrainMix | 0.352 | 0.469 |

Table 8: Results for TestModelAQ settings.

