# Peer review of "Learning through Dialogue Interactions by Asking Questions"

_ICLR 2017 — accepted_

[Author Response · Jiwei Li · 15 Dec 2016]
**Code/data release, paper update**

Dear all, 

We have released the data, code, and simulator described in the paper at

[Official Review · AnonReviewer4 · rating 7 · confidence 3 · 16 Dec 2016 (modified: 16 Jan 2017)]

The goal of this paper is to analyze the behaviour of dialogue agents when they must answer factoid questions, but must query an oracle for additional information. This can be interpreted as a form of interaction between the dialogue agent and a ‘teacher’.

The problem under investigation is indeed very important. The authors create a synthetic environment in which to test their agent. The main strength of the paper is that the paper tests many different combinations of environments, where either some knowledge is missing (and the agent has to query for it), or there is some misspelling in the teacher’s question, and different ways the agent can ask for extra information. 

I am a bit concerned that many of the tasks are too easy (e.g. the AQ question paraphrase), and I am also concerned that the environment presented is very limited, and quite far (in terms of richness of linguistic structure) from how real humans would interact with chatbots. I think the paper would be better positioned as testing the basic reasoning capabilities of agents/ their ability to do question answering, rather than dialogue. However, I think the ‘ground-up’ approach that starts with simple environments is indeed worthy of analysis, and this paper makes an interesting contribution in that direction. Of course, the paper would be much more convincing with human experiments. 

Additional notes:
I think the simulation, in the synthetic environment, for the first mistake a learner can make during dialogue: “the learner has problems understanding the surface form of the text of the dialogue partner, e.g., the phrasing of a question”, is particularly limited since only word misspellings are considered (and the models used don’t work at the character level), which is of course only a tiny fraction of ways an agent can misunderstand the context. I would be particularly interested to see some discussion of how the authors plan to scale this up to more realistic settings.

EDIT: I have updated my score to reflect the addition of the Mechanical Turk experiments

[Official Review · AnonReviewer1 · rating 8 · confidence 5 · 16 Dec 2016 (modified: 11 Jan 2017)]

This paper introduces a simulator and a set of synthetic tasks for evaluating a dialogue agent's ability to learn from user feedback. For solving these tasks, the paper uses memory networks (Sukhbaatar et al., 2015) learned through previously proposed supervised learning and reinforcement learning methods. In this setup, it is demonstrated that the agent learning from feedback (e.g. through question asking or question clarification) performs better.

The motivation for the paper is excellent; dialogue agents which learn directly from unstructured human feedback (as opposed to reward signals alone) could be very useful in real-world applications. However, the paper falls short on the execution. All the numerous experiments presented are based on the synthetic dialogue simulator, which is highly artificial and different from real-world dialogues. The simulator is based on a simple factoid question-answering framework, which normally is not considered dialogue and which appears to be solvable with a few hand-crafted rules. The framework also assumes that the user's feedback is always correct and is given in one of a handful of forms (e.g. paraphrase of original question without typos) and that the agent can learn from examples of another agent asking questions or making clarifications, which simplifies the task even further.

Because of the artificial setting and limited scope of the experiments, it seems difficult to draw conclusions about how to learn from unstructured user feedback. To test the hypothesis that it is possible to learn from such user feedback, I would strongly recommend the authors to continue working on this project by carrying out experiments with real human users (even in the factoid question answering domain, if necessary). This would provide much stronger evidence that a dialogue agent can learn from such feedback.


Other comments:
- The abstract uses the phrase "interactive dialogue agents". What is meant by "interactive" dialogue agents? All dialogue agents interact with the user, so isn't it redundant to call them interactive?
- A major limitation of the experiments is that the questions the agent can ask are specified a priori. If I understand correctly, in the supervised learning setting the agent is trained to imitate the questions of another rule-based agent. While in the RL setting, the paper states "For each dialogue, the bot takes two sequential actions $(a_1 , a_2)$: to ask or not to ask a question (denoted as a_1 ); and guessing the final answer (denoted as a_2)". This means the agent learns *when* to ask questions but not *what* questions to ask.
- Related to the previous comment, in the sub-section "ONLINE REINFORCEMENT LEARNING (RL)" the paper states "We also explored scenarios where the student learns the ability to decide when to ask a question and what to ask.". Please clarify this by removing the part "what to ask".
- The paper presents an overwhelming amount of results. I understand the benefit of synthetic tasks is precisely the ability to measure many aspects of model performance, but in this case it confuses the reader to present so many results. For example, what was the reason for including the "TrainAQ(+FP)" and "TrainMix" training settings? How do these results help validate the original hypothesis? If they don't, they should be taken out or moved to the appendix.
- Since the contribution of the paper lies in the tasks and evaluation, it might be better to move either the vanilla-MemN2N (Table 2) to the appendix or to move the Cont-MemN2N results (Table 3) to the appendix.

--- UPDATE ---

Following the discussion below and the additional experiments provided by the authors, I have increased my score to 8.

[Official Review · AnonReviewer3 · rating 7 · confidence 3 · 24 Dec 2016]

The paper introduces a simulator and a set of synthetic question answering tasks where interaction with the "teacher" via asking questions is desired. The motivation is that an intelligent agent can improve its performance by asking questions and getting corresponding feedback from users. The paper studies this problem in an offline supervised and an online reinforcement learning settings. The results show that the models improve by asking questions.  

-- The idea is novel, and is relatively unexplored in the research community. The paper serves as a good first step in that direction.
-- The paper studies three different types of tasks where the agent can benefit from user feedback.
-- The paper is well written and provides a clear and detailed description of the tasks, models and experimental settings.

Other comments/questions: 
-- What is the motivation behind using both vanilla-MemN2N AND Cont-MemN2N? Is using both resulting in any conclusions which are adding to the paper's contributions?
-- In the Question Clarification setting, what is the distribution of misspelled words over question entity, answer entity, relation entity or none of these? If most of the misspelled words come from relation entities, it might be a much easier problem than it seems.
-- The first point on Page 10 "The performance of TestModelAQ is worse than TestAQ but better than TestQA." is not true for Task 2 from the numbers in Tables 2 and 4.
-- What happens if the conversational history is smaller or none? 
-- Figure 5, Task 6, why does the accuracy for good student drop when it stops asking questions? It already knows the relevant facts, so asking questions is not providing any additional information to the good student. 
-- Figure 5, Task 2, the poor student is able to achieve almost 70% of the questions correct even without asking questions. I would expect this number to be quite low. Any explanation behind this?
-- Figure 1, Task 2 AQ, last sentence should have a negative response "(-)" instead of positive as currently shown. 

Preliminary Evaluation: 
A good first step in the research direction of learning dialogue agents from unstructured user interaction.

[Final Decision · Program Chairs · 06 Feb 2017]
**ICLR committee final decision**

This paper is a clear accept. Reviewers were both positive and confident about their assessments. Paper introduces a simulator and synthetic question answering task where interactions with the teacher are used for learning. Reviewers felt paper was well written with clear descriptions of tasks, models and experiments. Reviewer did comment on limitations due to the simple factoid QA framework explored for which hand crafted rules seems sufficient to solve the problem.